# Cross-Cultural Differences in the Perception of Lamb between New Zealand and Chinese Consumers in New Zealand

**DOI:** 10.3390/foods11142045

**Published:** 2022-07-11

**Authors:** Scott C. Hutchings, Luis Guerrero, Levi Smeets, Graham T. Eyres, Patrick Silcock, Enrique Pavan, Carolina E. Realini

**Affiliations:** 1AgResearch Limited, Te Ohu Rangahau Kai, Massey University Campus, Grasslands, Tennent Drive, Palmerston North 4474, New Zealand; scott.hutchings@agresearch.co.nz (S.C.H.); enrique.pavan@agresearch.co.nz (E.P.); 2IRTA-Monells, Finca Camps i Armet, 17121 Monells, Spain; lluis.guerrero@irta.cat; 3Department of Marketing & Supply Chain Management, Maastricht University, Nassaustraat 36, 5911 BV Venlo, The Netherlands; lgj.smeets@alumni.maastrichtuniversity.nl; 4Department of Food Science, University of Otago, Dunedin 9054, New Zealand; graham.eyres@otago.ac.nz (G.T.E.); pat.silcock@otago.ac.nz (P.S.); 5Departamento de Producción Animal, Estación Experimental Agropecuaria Balcarce, Instituto Nacional de Tecnología Agropecuaria, c.c. 276, Balcarce 7620, Argentina

**Keywords:** cross-cultural, preference, lamb, consumer, China, New Zealand

## Abstract

This study investigated differences between general New Zealand consumers and ethnic Chinese consumers living in New Zealand regarding the importance of lamb attributes at the point of purchase and opinions of New Zealand lamb. A central location test survey was undertaken with 156 New Zealand consumers living in Dunedin, New Zealand, and 159 Chinese consumers living in Auckland, New Zealand. In terms of importance at the point of purchase, Chinese consumers rated a number of attributes as more important than New Zealand consumers by a difference of >1.0 on a 9-point Likert scale for importance: animal origin, feeding, age, presence of hormones/residues, traceability, food safety, place of purchase, brand/quality label, and label information (*p* < 0.05). New Zealand consumers rated the price of other meats and animal welfare as more important than Chinese consumers (*p* < 0.05); however, the differences in scores were <1.0. In terms of opinions, Chinese consumers also considered New Zealand lamb to be better value for money, more additive-free, and more likely to make people feel good (*p* < 0.05), by scores >1.0 on a 7-point Likert scale for agreement. New Zealand consumers considered New Zealand lamb more traditional and boring (*p* < 0.05); however, the differences in scores were <1.0.

## 1. Introduction

In New Zealand, a nation with strong historical ties to lamb production [1], the consumption of lamb by local consumers has declined drastically in recent years. Lamb consumption in New Zealand is reported to have dropped 45% over the past decade [2], and total sheep meat consumption in New Zealand dropped from 25.5 kg per capita in 2002 to 3.5 kg per capita in 2019 [3,4]. In the meantime, the consumption of poultry and pork in New Zealand has increased [2,4], and veganism, vegetarianism, and flexitarianism are also on the rise [5].

The reasons for this decline in lamb consumption by New Zealand consumers is likely to be a result of a range of factors. Consumer concerns on issues such as the cost of lamb relative to other meats, the nutritive properties of lamb, environmental footprint of red meat production, the taste experience of lamb, perceived lack of versatility, as well as animal welfare, may all play a role in declining consumption. When considering lamb perception in other cultures, a wide range of attributes have been identified to be important to consumers when deciding to purchase lamb. For example, recent studies have shown that factors such as freshness, marbling, colour, the type of cut, price, and quality label are very important to lamb consumers in Brazil [6]; food safety and the absence of hormones and antibiotics are important for sheep meat consumers in Mexico [7]; and external appearance, origin, and price are important to lamb consumers in Spain [8].

While around 94% of the sheep meat produced in New Zealand is exported [9], an increasing desire by global consumers to buy local produce [10] and plans to reduce meat consumption in some key export markets due to environmental pressures [11,12], means that a greater understanding of the preferences of New Zealand consumers towards lamb is needed for New Zealand lamb producers to increase the share of product they market locally. Furthermore, understanding the attitudes of other growing ethnic populations in New Zealand, as well as general New Zealand consumers, can provide critical insights for New Zealand lamb producers on how to drive greater opportunities for local consumption of lamb. In New Zealand, Chinese are one of the fastest growing ethnic groups [13,14], largely as a result of immigration [15]. Understanding the perception of lamb of ethnic Chinese in New Zealand provides an opportunity to understand cross-cultural differences in perception towards lamb, from consumers whose culture originates from a nation where lamb consumption has been increasing in recent decades [16].

Prior cross-cultural research has identified notable differences in how consumers from different cultures view lamb. For example, Indian and Chinese consumers have been shown to have a higher willingness to pay for environmental certification on lamb products than UK consumers [17]. A number of studies have also investigated differences in sensory/eating quality responses of lamb between Chinese consumers and consumers in certain western markets [18,19,20]. While the attitudes of Chinese consumer perceptions in China towards lamb are well understood [21,22,23], to the best of the authors’ knowledge, no published research has investigated attitudes of New Zealand consumers towards a comprehensive range of credence and intrinsic New Zealand lamb attributes, or investigated the attitudes of any specific New Zealand ethnic group.

The aim of this study was, therefore, to determine how New Zealand consumers and Chinese consumers in New Zealand differ in their attitudes towards a range of lamb attributes (such as animal origin, food safety, appearance, taste, price, brand), and in their opinions of New Zealand lamb. It was hypothesized that the relative importance of a range of attributes, and opinion of New Zealand lamb would differ between the cultures.

## 2. Materials and Methods

### 2.1. Recruitment, Sample Characteristics and Data Collection

A sample of 156 general New Zealand consumers living in Dunedin, New Zealand, was obtained through a consumer database at the University of Otago, and a sample of 159 ethnic Chinese consumers living in Auckland, New Zealand, was obtained for the Chinese consumer group by an external recruitment company. All Chinese consumers were self-identified Chinese, all spoke Mandarin as their first language, and read/wrote in Mandarin script. 

The survey was administered to the New Zealand consumers at the Department of Food Science, the University of Otago, Dunedin, and to the Chinese consumers at the Plant & Food Research Consumer Research facility in Auckland. Eight sessions with 20 participants in each session were run at central location facilities for both groups of consumers. The survey was undertaken in central location facilities, rather than through an online survey, as studies using Likert scales in online surveys with Chinese consumers can often show that attribute differentiation can be limited, where Chinese consumers give high scores for all attributes, using a narrow scale range [21,24]. Some studies suggest that the behavioural differences between consumers when completing surveys in person, compared to online, may increase scale use [25] and the reliability of results [26].

Both groups of consumers received and completed the questionnaire on paper ballots, in individual sensory booths in January, 2019. A summary of the demographic characteristics of the two population groups is shown in Table 1. All consumers were non-rejecters of lamb and, on average, consumed lamb at least once per month, and were aged 18–75 years old. The data for the New Zealand consumers were collected and stored in accordance with University of Otago Human Ethics application number 15/092. The data for the Chinese consumers were covered by general approval for sensory and consumer research from the Human Ethics Committee at the New Zealand Institute for Plant and Food Research. Participants provided informed consent and were assured that their responses would remain confidential and that they could withdraw from the study at any time.

### 2.2. Questionnaire

The questionnaire used in this survey, conducted as central location test, asked identical questions to the online survey conducted in China and results published in [21]. A copy of the questionnaire can be found in Appendix A. The survey captured information on demographic data, dietary habits, meat qualities of interest at the point of purchase, and the lamb products the consumers typically purchase. To understand consumer considerations at the point of purchase, consumers indicated the level of importance of varying aspects of lamb meat purchase on a scale of one (“not important”) to nine (“very important”). This included animal and other production factors, pricing factors, intrinsic cues of the meat, convenience factors, personal knowledge of commercial cuts and aspects related to descriptive information and branding. To capture their opinion on New Zealand lamb meat, consumers rated the degree of agreement on several descriptions of the lamb meat on a scale of one (“strongly disagree”) to seven (“strongly agree”). These describe New Zealand lamb meat in several ways, including, but not limited to, being nutritious, safe, good value for money, produced sustainably and convenient. All New Zealand consumers received the English version of the questionnaire, while Chinese consumers received the Mandarin version of the questionnaire (Appendix A). A couple of native speakers of Mandarin translated the questionnaire from English into Mandarin. Chinese scientists in Auckland (Plant and Food Research, New Zealand) subsequently validated the translation.

### 2.3. Data Analysis

XLSTAT 2017 (Addinsoft 2012) software was used to analyze survey data. For data on demographic factors, diet, and consumption patterns, a Chi-square test was performed to find differences between ethnicities. A one-way analysis of variance (ANOVA) was applied to the data on consumers’ consideration of lamb meat attributes at the point of purchase and their opinion on New Zealand lamb meat to find differences in scores between ethnicities (fixed effect).

## 3. Results

### 3.1. Demographic Characteristics

Table 1 summarizes the demographic characteristics of the two consumer groups that were sampled. The general New Zealand consumer group was comprised of a majority of New Zealand Europeans (87%), with smaller proportions of other ethnicities (such as Maori, Samaon, Cook Island Maori, and Chinese), while the Chinese group was comprised of only people of Chinese ethnicity (100%). Gender balance was similar between the two groups, while age distribution did differ slightly—more of the Chinese consumers were aged 26–35, and more of the New Zealand consumers were aged 46–60. Educational background also differed, with a higher proportion of Chinese consumers having a tertiary qualification, and a higher proportion of New Zealand consumers having a trades certificate or only a high school qualification. New Zealand consumers and Chinese consumers were similar in terms of occupation types, apart from a slightly higher number of New Zealand consumers working as trades people and a lower number as home makers. Income was also similar between the two groups, with the exception of a higher proportion of Chinese consumers earning NZD40,001 to NZD55,000, and a higher proportion of New Zealand consumers earning more than NZD150,000. The number of adults in the household was similar between the two groups; however, having 1 child in the household was slightly more common for Chinese consumers, while having 2 or more children in the household was slightly more common for New Zealand consumers.

### 3.2. Diet and Consumption Patterns

Results from Table 2 show that diet differed significantly between New Zealand and Chinese consumers, with a much greater proportion of Chinese consumers following low salt, low sugar, and low calorie diets (*p* < 0.05). In terms of consumption frequency of animal protein sources, Chinese consumers generally consumed significantly greater quantities of lamb and pork than New Zealand consumers (*p* < 0.05), and significantly less poultry and beef (*p* < 0.05). No significant differences in fish consumption were found between the two groups (*p* > 0.05) (Table 2).

### 3.3. Preferred Level of Cooking, Meat Qualities of Interest, Purchase Location and Types of Lamb Products Typically Purchased

Many of the preferences measured in terms of cooking, location of purchase, and types of lamb products differed significantly between New Zealand and Chinese consumers (*p* < 0.05) (Table 3). A higher proportion of Chinese consumers preferred meat cooked well-done, while a higher percentage of New Zealand consumers preferred meat cooked rare (*p* < 0.05). A higher proportion of Chinese consumers sought out leanness and meat colour when purchasing meat, whereas a higher proportion of New Zealand consumers were interested in portion size (*p* < 0.05). No difference in the proportion of consumers looking at price when purchasing red meat was observed (*p* > 0.05). Although the proportion of Chinese consumers that look for marbling when purchasing lamb was higher (*p* < 0.05) than New Zealand consumers, the proportion of consumers looking for marbling was relatively low for both ethnicities. Similar proportions of New Zealand and Chinese consumers typically purchased leg roast lamb. A greater proportion of New Zealand consumers typically purchased lamb chops, lamb mince, and lamb sausages, whereas a greater proportion of Chinese consumers typically purchased lamb steaks, lamb shanks, and lamb shoulder roast (*p* < 0.05).

### 3.4. Importance of Lamb Attributes at the Point of Purchase (e.g., Origin, Food Safety, Appearance, Taste, Price)

In general, both New Zealand consumers and Chinese consumers rated most lamb attributes at the point of purchase as important (scores > 5.0). Most notably, both New Zealand consumers and Chinese consumers rated taste attributes (flavour and texture in particular), food safety, and price, as important lamb attributes at the point of purchase. Furthermore, both New Zealand consumers and Chinese consumers also rated animal sex and the time of day as relatively unimportant attributes at the point of purchase (scores < 5.0) (Table 4). However, comparisons between New Zealand consumers and Chinese consumers for each attribute revealed significant differences for 20 of the 25 attributes (*p* < 0.05) (Table 4). While the magnitude of some differences was small (differences in scores <1.0), some notable effects were observed: Chinese consumers rated animal origin, feeding, age, presence of hormones/residues, traceability, food safety, place of purchase, brand/quality label, and label information as more important than New Zealand consumers by a difference of 1.0 or more (*p* < 0.05). New Zealand consumers rated animal welfare and the price of other meats as significantly more important than Chinese consumers (*p* < 0.05), although the magnitude of differences with Chinese consumers was less than one. There were no significant differences between New Zealand consumers and Chinese consumers in their ratings of the importance of lamb price, fat content, trust in the butcher, dish to be prepared with and value for money (*p* > 0.05).

### 3.5. Opinion about New Zealand Lamb

In general, both New Zealand consumers and Chinese consumers agreed with the positive statements presented on New Zealand lamb characteristics (scores > 4.0), and also both showed disagreement with the two negative statements presented on New Zealand lamb characteristics (scores < 4.0), boring and hard to digest (Table 5). However, there were also significant differences between New Zealand consumers and Chinese consumers in the degree of agreement/disagreement about New Zealand lamb for 15 out of 18 characteristics (*p* < 0.05). Specifically, Chinese consumers’ opinion of New Zealand lamb was higher than New Zealand consumers for the following characteristics: nutritious, healthy, safe, good value for money, natural, produced sustainably, convenient, readily available, high quality, contains no additives, makes people feel good, tastes good, and supports the New Zealand economy (*p* < 0.05). However, the magnitude of the differences was small in most cases (differences in scores < 1.0), except for the attributes good value for money, makes people feel good, and contains no additive (differences in scores > 1.0). Interestingly, New Zealand consumers’ opinion of New Zealand lamb was higher than Chinese consumers for two characteristics: traditional product and boring (*p* < 0.05), although the magnitude of the differences was less than one. There was no significant difference between the opinions of New Zealand consumers and Chinese consumers for well-known, unique, and hard to digest (*p* > 0.05).

## 4. Discussion

### 4.1. Differences in Diets, Cooking and Purchasing Habits between Chinese and New Zealand Consumers

As expected, results showed a number of differences between Chinese consumers and New Zealand consumers in their diet and purchasing behaviour with meat (Table 2). This may occur as a result of differences in a wide range of social (e.g., cultural traditions, religion, family style), economic (e.g., salary, lifestyle or employment status), and environmental (e.g., water, soil, geography) factors that New Zealand consumers and Chinese consumers would have experienced growing up [27]. In general, Chinese consumers are well-known to be health-conscious consumers [28], and the present study’s finding that a significantly higher proportion of Chinese consumers in New Zealand followed diets low in salt, sugar, and calories, supports this concept. Although results indicated that a higher proportion of Chinese consumers look for marbling when purchasing lamb compared to New Zealand consumers, results also indicate a relatively low proportion of consumers from both ethnicities look for marbling when purchasing lamb compared to most other quality attributes. Large differences in the types of lamb products purchased were also seen between Chinese and New Zealand consumers, which again was expected as a result of differences in food culture such as food familiarity, flavour preferences, and cooking styles [18,20,27,29]. Chinese consumers tended to prefer to cook their lamb more thoroughly than New Zealand consumers, which may again reflect differences in cooking styles between cultures. For example, the most common cooking methods for sheep meat in China are stewing, cooking in hot pot or roasting [23], which normally cook meat to a high degree of doneness.

### 4.2. Differences in the Importance of Lamb Attributes at the Point of Purchase between Chinese and New Zealand Consumers

It is well-known that Chinese consumers place high importance on the quality, safety, and traceability of meat [22,23,30], often seeing these attributes as more important than consumers in other western nations [17]. Results in this study support this notion, as higher ratings of importance for food safety-related attributes such as traceability, brand/quality label, label information, food safety, knowledge of commercial cuts and place of purchase were found for Chinese consumers compared to New Zealand consumers. Several animal factors (origin, feed, and age) were also more important to Chinese consumers than New Zealand consumers (Table 4). The higher value of importance of animal attributes by Chinese consumers may stem from a high level of distrust in Chinese food production systems and Chinese food products [31,32]. Animal welfare was an exception, where New Zealand consumers considered it more important than Chinese consumers when purchasing lamb. Recent studies suggest that animal welfare is becoming an increasingly important issue with Chinese consumers [33,34,35]; however, the results of this study indicate that it may still be a more important issue for consumers from a western background. In New Zealand, consumers generally expect the food they purchase to be safe for consumption; this high degree of trust may lead to greater attention on animal welfare issues.

In terms of the importance of sensory attributes, while Chinese consumers gave slightly higher overall ratings than New Zealand consumers (meat appearance, colour, flavour, texture), both consumer groups considered these attributes important when purchasing lamb (Table 4). In fact, meat flavour and texture were of a comparative level of importance to food safety for New Zealand consumers (food safety was more important than flavour and texture for Chinese consumers). Previous studies assessing the importance of sensory attributes on food choice suggest that sensory appeal is important to both Chinese and western consumers [36]. Interestingly, Prescott et al., 2002, [37] compared motives of food choice between New Zealand consumers and Japanese, Taiwanese, and ethnically Chinese Malaysian consumers, and found that sensory appeal was more important for New Zealand consumers than consumers from the other nations.

### 4.3. Differences in Opinion of New Zealand Lamb between Chinese and New Zealand Consumers

The largest difference observed between New Zealand consumers and Chinese consumers in the opinion of New Zealand lamb was around price, where Chinese consumers considered New Zealand lamb as much better value for money than New Zealand consumers (Table 5). This result may again be due to the high value Chinese consumers place on food safety and quality attributes in lamb, attributes which the results of this study suggest Chinese consumers see in New Zealand lamb (Table 5). The fact that New Zealand consumers are known to view the price of food as the most important attribute when making purchasing decisions [38], is also likely to contribute to this result. New Zealand consumers rated the price of other meats as more important than Chinese consumers at the point of purchase (Table 4), and New Zealand consumers’ familiarity with New Zealand lamb, where safety and quality are expected, may temper concerns for some other attributes and put greater emphasis on price.

Furthermore, Chinese consumers had a higher overall opinion of New Zealand lamb than New Zealand consumers (Table 5). Although the magnitude of differences was relatively small, results suggest that Chinese consumers considered New Zealand lamb as more additive-free, likely to make people feel good, natural, healthy, sustainable, and high quality than New Zealand consumers; whereas New Zealand consumers viewed New Zealand lamb as more traditional than Chinese consumers (Table 5). In addition, Chinese consumers disagreed even more than New Zealand consumers with the statement that New Zealand lamb is boring. These results suggest that the New Zealand lamb industry has a significant opportunity to market lamb and lamb-based products to Chinese living in New Zealand, but also highlights that strategies need to be developed to address the concerns of general New Zealand consumers that New Zealand lamb is too expensive. While results in this study suggest that New Zealand consumers continue to believe that New Zealand lamb is unique, well-known, and of high quality, it is interesting that New Zealand consumers appear to be largely indifferent towards New Zealand lamb attributes despite its strong international reputation [1,39].

### 4.4. Limitations of This Study

This study recruited participants who were diverse in terms of sex, age, educational background, occupation, income, and household size. However, the age distribution of the New Zealand sample was skewed slightly towards the 18–25- and 46–60-year-old age categories, and skewed slightly towards males, compared to the total New Zealand population [40]. The age distribution of the Chinese sample was skewed slightly towards the younger/middle age categories (26–45 years old), and towards females, compared to the total Chinese population in New Zealand [41].

The results of this study show cross-cultural differences between New Zealanders and Chinese living in New Zealand; however, results cannot be generalized to cross-cultural differences between New Zealand consumers and Chinese consumers from China. A comparison of responses of Chinese consumers from this study, with those obtained from an online survey with Chinese consumers in China using identical questions [21], shows only moderate differences in the results.

## 5. Conclusions

This study showed that, in terms of importance at the point of purchase, Chinese consumers rated a number of attributes as more important than New Zealand consumers by a difference of >1.0 on a 9-point Likert scale for importance: animal origin, feeding, age, presence of hormones/residues, traceability, food safety, place of purchase, brand/quality label, and label information (*p* < 0.05). New Zealand consumers rated the price of other meats and animal welfare as more important than Chinese consumers (*p* < 0.05); however, the differences in scores were <1.0. Both Chinese and New Zealand consumers rated sensory properties of lamb (meat colour, meat appearance, meat flavour, and meat texture) as important relative to other attributes; however, Chinese consumers still rated sensory properties as significantly more important than New Zealand consumers (*p* < 0.05), despite the differences in scores being <1.0.

In terms of opinions, Chinese consumers also considered New Zealand lamb to be better value for money, more additive-free, and more likely to make people feel good (*p* < 0.05), by scores >1.0 on a 7-point Likert scale for agreement. New Zealand consumers considered New Zealand lamb more traditional and boring (*p* < 0.05); however, the differences in scores were <1.0.

These insights may prove valuable for the New Zealand lamb industry to develop strategies to address declining domestic lamb consumption in New Zealand.

## Figures and Tables

**Table 1 foods-11-02045-t001:** Demographic characteristics of New Zealand and Chinese consumers in New Zealand (%).

	NZ	CN-NZ	*p* (Chi^2^)
*Ethnicity*			
NZ European	86.6	0.0	
Maori	1.9	0.0	
Samoan	0.5	0.0	
Cook Island Maori	0.5	0.0	
Tongan	0.0	0.0	
Niuean	0.0	0.0	
Chinese	3.8	100.0	
Indian	0.5	0.0	
Other	6.2	0.0	
*Gender*			
Male	53.8	43.7	0.071
Female	46.2	56.3	0.072
*Age*			
18–25	23.7	15.8	0.079
26–35	14.7	43.0	<0.001
36–45	15.4	24.1	0.054
46–60	31.4	8.9	<0.001
61 and over	14.1	8.2	0.098
*Education*			
Tier 1 *	3.9	0.0	0.012
Tier 2 ^#^	4.5	0.0	0.007
Tier 3 ^@^	12.3	1.3	<0.001
Tier 4 ^^^	17.5	5.1	0.001
Tier 5 ^$^	13.0	5.7	0.027
Tier 6 ^+^	48.1	88.0	<0.001
*Occupation*			
Trades	10.5	3.8	0.022
Professional	27.5	32.3	0.353
Administration/Office	5.2	1.9	0.112
Sales/Services	7.8	7.0	0.767
Technical	6.5	3.8	0.274
Labourer	1.3	0.0	0.149
Home maker	0.7	5.7	0.012
Student	20.3	28.5	0.092
Retired	9.2	7.6	0.620
Unemployed	1.3	0.6	0.543
Other employment	9.8	8.9	0.775
*Income*			
Less than NZD25,000	14.5	9.6	0.190
NZD25,001 to NZD40,000	17.1	12.8	0.292
NZD40,001 to NZD55,000	6.6	19.9	<0.001
NZD55,001 to NZD70,000	7.8	16.0	0.028
NZD70,001 to NZD100,000	23.7	22.4	0.795
NZD100,001 to NZD150,000	16.4	14.1	0.567
More than NZD150,000	13.8	5.1	0.009
*Adults in household*			
1	8.4	12.2	0.271
2	59.4	53.2	0.274
3	22.6	19.9	0.559
4 or more	9.7	14.7	0.173
*Children in household*			
0	64.1	68.6	0.403
1	10.3	18.9	0.030
2 or more	25.6	12.6	0.003

NZ (New Zealand); CN-NZ (Chinese consumers in New Zealand); * None; ^#^ Primary school; ^@^ Middle school; ^^^ High school; ^$^ Trades certificate or vocational college; ^+^ Bachelor’s degree or higher.

**Table 2 foods-11-02045-t002:** Dietary restrictions and consumption frequency of animal protein sources (%) (*p* value determined using a Chi-squared test for ethnicity).

	NZ	CN-NZ	*p* (Chi^2^)
*Dietary Restrictions*			
Low salt	3.2	62.3	<0.001
Low sugar	5.8	62.9	<0.001
Low calories	2.6	42.1	<0.001
Do not follow diets	88.5	24.5	<0.001
*Lamb*			
Daily	0.0	1.3	0.160
4–5 times a week	0.6	0.0	0.312
2–3 times a week	4.5	15.3	0.002
Weekly	19.5	31.8	0.013
Fortnightly	35.1	28.0	0.182
Monthly	40.3	23.6	0.002
Never		-	-
*Beef*			
Daily	0.6	3.2	0.106
4–5 times a week	9.1	4.4	0.100
2–3 times a week	33.8	20.9	0.011
Weekly	40.3	41.1	0.874
Fortnightly	11.7	20.3	0.039
Monthly	3.2	9.5	0.024
Never	1.3	0.6	0.547
*Pork*			
Daily	0.0	3.8	0.017
4–5 times a week	1.4	8.2	0.006
2–3 times a week	9.6	13.3	0.312
Weekly	27.4	27.8	0.930
Fortnightly	28.1	14.6	0.004
Monthly	28.8	22.2	0.185
Never	4.8	10.1	0.079
*Poultry*			
Daily	1.3	0.6	0.547
4–5 times a week	9.9	9.0	0.788
2–3 times a week	43.4	25.0	0.001
Weekly	36.2	36.5	0.948
Fortnightly	5.9	10.3	0.164
Monthly	2.6	12.2	0.001
Never	0.7	6.4	0.007
*Fish*			
Daily	0.7	1.9	0.343
4–5 times a week	2.0	2.5	0.761
2–3 times a week	12.8	19.6	0.103
Weekly	28.9	26.6	0.656
Fortnightly	22.8	21.5	0.784
Monthly	27.5	25.9	0.756
Never	5.4	1.9	0.102

NZ (New Zealand); CN-NZ (Chinese consumers in New Zealand).

**Table 3 foods-11-02045-t003:** Preferred level of cooking, meat qualities of interest to consumers at the point of purchase and purchase frequency of different lamb products (%) (*p* value determined using a Fishers exact test (Chi-squared) for ethnicity).

	NZ	CN-NZ	*p* (Chi^2^)
*Preferred Level of Cooking*			
Rare	33.5	3.8	<0.001
Medium/Rare	5.8	9.5	0.220
Medium	18.7	19.6	0.838
Medium/Well-done	38.7	49.4	0.058
Well-done	3.2	17.7	<0.001
*What qualities do you look for when* *purchasing red meat?*			
Marbling	23.7	45.9	<0.001
Leanness	44.9	83.0	<0.001
Meat Colour	49.4	91.2	<0.001
Portion size	62.8	34.6	<0.001
Price	85.3	80.5	0.263
*What lamb products do you typically purchase?*			
Leg Roast	58.3	61.6	0.550
Lamb chops	71.2	42.8	<0.001
Lamb mince	26.9	13.2	0.002
Lamb rump	5.1	8.2	0.278
Lamb steaks	39.1	53.5	0.011
Lamb sausages	34.6	11.3	<0.001
Lamb shanks	28.8	40.3	0.033
Shoulder roast	28.8	42.8	0.010

NZ (New Zealand); CN-NZ (Chinese consumers in New Zealand).

**Table 4 foods-11-02045-t004:** The relative importance of lamb attributes at the point of purchase for New Zealand consumers and Chinese consumers (mean ± SD) (1 = not important, 9 = very important) (*p* value determined using ANOVA with ethnicity as main effect).

	NZ	SD	CN-NZ	SD	*p* (ANOVA)
Animal origin	5.308	2.549	6.513	2.332	<0.001
Animal welfare	5.871	2.349	5.095	2.431	0.004
Animal feeding	5.474	2.185	6.778	1.934	<0.001
Animal age	4.660	2.231	6.228	2.203	<0.001
Animal sex	2.232	1.840	3.513	2.214	<0.001
Pres. of hormones/residues	6.141	2.541	8.063	1.440	<0.001
Traceability	4.923	2.579	6.006	2.325	<0.001
Lamb price	7.277	1.700	6.981	1.725	0.127
Price of other meats	6.768	2.032	6.044	2.166	0.003
Fat content	6.542	1.821	6.310	2.099	0.299
Meat appearance	7.000	1.381	7.861	1.068	<0.001
Meat colour	6.908	1.607	7.943	1.048	<0.001
Meat flavour	7.559	1.404	8.000	1.162	0.003
Meat texture (tenderness)	7.503	1.333	7.943	1.130	0.002
Food safety (risk of disease)	7.497	2.314	8.842	0.511	<0.001
Place of purchase	4.922	2.246	6.582	1.939	<0.001
Trust in butcher	5.806	2.468	5.633	2.201	0.512
Time of day to purchase	2.613	2.037	4.741	2.630	<0.001
Brand or quality label	4.686	2.424	7.151	1.726	<0.001
Label information	5.574	2.253	6.791	1.896	<0.001
Presentation (piece/slice/etc.)	5.753	2.270	6.854	1.583	<0.001
Ease of preparation	6.071	2.095	6.987	1.806	<0.001
Dish to be prepared with it	5.374	2.154	5.810	2.224	0.079
Knowledge of commercial cuts	5.118	2.137	6.354	1.919	<0.001
Value for money	7.315	1.628	7.405	1.584	0.625

Rated 1–9. NZ (New Zealand); CN-NZ (Chinese consumers in New Zealand).

**Table 5 foods-11-02045-t005:** New Zealand consumers and Chinese consumers’ opinion of New Zealand lamb (1 = strongly disagree, 7 = strongly agree) (mean ± SD) (*p* value determined using ANOVA with ethnicity as main effect).

	NZ	SD	CN-NZ	SD	*p* (ANOVA)
Nutritious	5.955	1.044	6.229	0.831	0.011
Healthy	5.921	1.064	6.459	0.738	<0.001
Well-known	6.201	1.168	6.395	0.979	0.114
Unique	5.020	1.440	5.000	1.476	0.905
Safe	6.059	1.015	6.389	0.748	0.001
Good value for money	4.292	1.440	5.968	1.087	<0.001
Boring	2.513	1.540	1.898	1.199	<0.001
Traditional product	5.575	1.271	4.809	1.661	<0.001
Natural	5.523	1.298	6.306	0.998	<0.001
Hard to digest	2.477	1.410	2.293	1.424	0.254
Produced sustainably	4.941	1.368	5.468	1.277	0.001
Convenient	4.914	1.332	5.369	1.247	0.002
Readily available	5.448	1.391	6.168	1.022	<0.001
High quality	5.714	1.203	6.287	0.801	<0.001
Contains no additive	4.843	1.410	6.032	1.112	<0.001
Makes people feel good	5.072	1.257	6.255	0.800	<0.001
Taste good	6.091	0.986	6.306	0.790	0.035
Supports NZ economy	5.877	1.345	6.497	0.874	<0.001

Rated 1–7. NZ (New Zealand); CN-NZ (Chinese consumers in New Zealand).

## Data Availability

The data presented in this study are available on request from the corresponding author. Although consumer data have been anonymised, data are not publicly available.

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
