# Peer review of "Cross-Cultural Differences in the Perception of Lamb between New Zealand and Chinese Consumers in New Zealand"

_foods, 2022, doi:10.3390/foods11142045_

Round 1
Reviewer 1 Report
Cross-cultural differences in the perception of lamb between New Zealand and Chinese consumers in New Zealand
This work represents a relevant contribution to understanding which attributes are associated with lamb consumption in New Zealand. Also, the results of this work may be valuable for the development of strategies to reverse the decline in domestic sheep meat consumption in New Zealand.
The article is well organized and clearly written, and as such, it is simple to follow. Also, results are supported by methodologies with scientific correctness. The Tables are relevant for understanding the article. The results are well discussed with the existing knowledge on the subject and support the conclusions.
Some detailed comments are below:
Please check the Table 4 head. It is different from the other tables. The same for Table 5
Please check the Table 4 head. Está diferente das outras tableas. The same for Table 5
L239 Please check “ 3.5 Opinion of New Zealand Lamb” it is suggested to change to “3.5 Opinion about New Zealand Lamb”
Author Response
Comments and Suggestions for Authors
Cross-cultural differences in the perception of lamb between New Zealand and Chinese consumers in New Zealand
This work represents a relevant contribution to understanding which attributes are associated with lamb consumption in New Zealand. Also, the results of this work may be valuable for the development of strategies to reverse the decline in domestic sheep meat consumption in New Zealand.
The article is well organized and clearly written, and as such, it is simple to follow. Also, results are supported by methodologies with scientific correctness. The Tables are relevant for understanding the article. The results are well discussed with the existing knowledge on the subject and support the conclusions.
Some detailed comments are below:
Please check the Table 4 head. It is different from the other tables. The same for Table 5
Table headers have been edited to ensure consistency.
L239 Please check “ 3.5 Opinion of New Zealand Lamb” it is suggested to change to “3.5 Opinion about New Zealand Lamb”
Change made.
Reviewer 2 Report
The manuscript titled Cross-cultural differences in the perception of lamb between New Zealand and Chinese consumers in New Zealand investigates if two groups of consumers living in New Zealand differ in attitudes concerning lamb. The article is interesting. The methodology is appropriate, whereas arguments have been developed on an appropriate theoretical background. The article is generally well written and in my opinion it is needs just a Minor Revision. Following shortcomings needs to be corrected:
1. Line 54: Please define NZ abbreviation when first time used.
2. Point 4.4: Please present an information on the sample representativeness.
Author Response
Comments and Suggestions for Authors
The manuscript titled Cross-cultural differences in the perception of lamb between New Zealand and Chinese consumers in New Zealand investigates if two groups of consumers living in New Zealand differ in attitudes concerning lamb. The article is interesting. The methodology is appropriate, whereas arguments have been developed on an appropriate theoretical background. The article is generally well written and in my opinion it is needs just a Minor Revision. Following shortcomings needs to be corrected:
- Line 54: Please define NZ abbreviation when first time used.
The text has been edited to replace NZ with New Zealand throughout the manuscript (with the exception of the Tables, where the NZ abbreviation is already defined). The graphical abstract has also been edited to define the NZ abbreviation.
- Point 4.4: Please present an information on the sample representativeness.
The following paragraph has been added to point 4.4 to address sample representativeness:
This study recruited participants who were diverse in terms of sex, age, educational background, occupation, income, and household size. However, the age distribution of the New Zealand sample was skewed slightly towards the 18-25 and 46–60 year-old age categories, and skewed slightly towards males, compared to the total New Zealand population [40]. The age distribution of the Chinese sample was also skewed slightly towards the younger/middle age categories (26-45 years old), and towards females, compared to the total Chinese population in New Zealand [41].